# The Slx4-Rad1-Rad10 nuclease differentially regulates deletions and duplications induced by a replication fork barrier

Marina K Triplett [1,2], Iffat Ahmed [2], Swathi Shekharan [3], Lorraine S Symington [2,4*]

**1** Integrated Program in Cellular, Molecular, and Biomedical Studies, Columbia University Irving Medical Center, New York, New York, United States of America, **2** Department of Microbiology & Immunology, Columbia University Irving Medical Center, New York, New York, United States of America, **3** Columbia University/Amgen Summer Undergraduate Research Program, New York, New York, United States of America, **4** Department of Genetics & Development, Columbia University Irving Medical Center, New York, New York, United States of America

* lss5@cumc.columbia.edu

## Abstract

Genome instability is a hallmark of cancer that can be caused by DNA replication stress. Copy number variation (CNV) is a type of genomic instability that has been associated with both tumorigenesis and drug resistance, but how these structural variants form in response to replication stress is not fully understood. Here, we established a direct repeat genetic reporter in *Saccharomyces cerevisiae* to detect recombination events that result in either a duplication or a deletion. Using this system, we measured recombination resulting from site-specific replication fork stalling initiated by Tus binding to an array of *Ter* sites. We found that a Tus/*Ter* fork block downstream of direct repeats induced CNV by a mechanism involving the Mph1 translocase, Exo1-catalyzed end resection and Rad51-dependent strand invasion. While the Slx4 scaffold protein and its nuclease-binding partner, Rad1-Rad10, were shown to be required for duplications, we found that they suppress deletion formation in this context. These opposing functions suggest that both recombination products arise through a large loop heteroduplex intermediate that is cleaved by Rad1-Rad10 in a manner that promotes duplications and eliminates deletions. Taken together, these studies give insight into the mechanisms governing CNV in the context of replication fork stalling, which may ultimately provide a better understanding of how replication stress contributes to cancer and other diseases characterized by genome instability.

## Author summary

Genome instability is characterized by changes in the identity, number, or location of DNA sequences in the genome and has been associated with cancer, aging, and developmental disorders. DNA replication stress is one of the various

**Data availability statement:** The authors confirm that all data underlying the findings are fully available without restriction. All relevant data are within the paper and its Supporting Information files.

**Funding:** This study was supported by funding from the National Institutes of Health, USA, grants R35 GM126997 and P01 CA174653 to L.S.S., and by the NSF Graduate Research Fellowship number DGE 2036197 to M.K.T. The funders had no role in study design, data collection and analysis, decision to publish, or preparation of the manuscript.

**Competing interests:** The authors have declared that no competing interests exist

causes of genome instability and has been shown to promote copy number variations (CNVs); however, the exact mechanism of how CNVs arise in response to disruption of replication remains unclear. To study the relationship between replication stress and CNVs, we designed a genetic assay in budding yeast to detect duplications and deletions and measured their frequencies in response to replication fork stalling at a specific genomic site. We found that both duplications and deletions are stimulated by a stalled replication fork and that these CNVs are promoted or suppressed by specific genes. Our findings provide a potential mechanism for copy number gain or loss in response to a stalled fork, which could ultimately help us understand how replication stress can lead to genome instability associated with cancer and other genomic disorders.

## Introduction

Maintaining genome integrity requires accurate and complete duplication of the genome in each cell cycle. In all organisms, the progression of replication forks is challenged by both endogenous and exogenous stresses. Endogenous sources of replication stress include DNA lesions, protein-DNA barriers, nucleotide depletion, DNA secondary structures, and transcription-replication conflicts [1]. These obstacles can hinder proper replication fork progression and lead to replication fork stalling or collapse [1]. Replication failure caused by global replication stress or replication fork barriers (RFBs) is a source of genomic instability in cancer and developmental disorders [2].

There are several ways in which DNA replication forks can be restarted or rescued in response to stress, including dormant origin firing, repriming, and translesion synthesis, in addition to mechanisms involving homologous recombination (HR) [1,3]. One of these homology-mediated pathways is template switching, in which the strand undergoing synthesis switches to the sister chromatid to bypass the lesion and continue replication. Another is fork reversal, which involves the reannealing of the parental strands and annealing of the nascent strands to form a four-way junction that resembles a Holliday junction [4–6]. The end of this reversed fork structure could undergo resection and invade the parental strands to resume replication, or the structure could be cleaved by nucleases to form a one-ended double-strand break (DSB) that would then use the sister chromatid to template repair [3,4,7]. While recombination-dependent mechanisms of replication fork restart are mostly error-free, they can also lead to genomic rearrangements and genomic instability if recombination occurs at an ectopic location [8,9].

Repetitive sequences are abundant in eukaryotic genomes and have the potential to give rise to structural variation via homology-dependent mechanisms. Repeats can be arranged in direct or inverted orientation, or they can be dispersed throughout the genome. Non-allelic HR between repeats can result in loss or gain of sequences (copy number variation), inversions or chromosome translocations, and such events are associated with genomic disorders [10–12]. Gene amplification has also been

linked to drug resistance in a variety of systems and can contribute to adaptation to environmental stress [13–16]. Studies in *Saccharomyces cerevisiae* have shown that replication fork stalling in response to high levels of transcription or a protein barrier can drive copy number variation (CNV) of the *CUP1* tandem array [17,18], linking replication stress to repeat instability. Furthermore, whole genome sequencing studies in humans have found that some forms of cancer exhibit specific patterns of structural variation that have the potential to serve as genomic biomarkers [19–21]. One example of this type of signature is the tandem duplicator (TD) phenotype, which is characterized by large numbers of repetitive, adjacent DNA sequences distributed across the genome and is enriched in breast and ovarian cancer genomes [19]. Short-span TDs of ~10 kb are typically found in BRCA1$^{-/-}$ but not BRCA2$^{-/-}$ tumors [19,20,22], and have been linked to replication stress [22–26].

Replication stress has often been studied using chemicals that slow or stall replication forks, the most common being hydroxyurea (which depletes dNTPs) and aphidicolin (a DNA polymerase inhibitor) [27,28]. These methods, when combined with genetic recombination reporter assays, have been useful in elucidating replication stress-induced recombination mechanisms on a global level. However, these approaches also have limitations in that the number and location of DNA damage/fork stalling sites cannot be controlled. The development of site-specific methods of inducing DNA replication fork stalling or collapse, such as the *RTS1* barrier in *Schizosaccharomyces pombe* and the bacterial Tus/*Ter* barrier that has been applied in eukaryotes [9,29–31], have provided opportunities for replication stress-induced recombination to be studied at single locations in the genome.

The Tus/*Ter* system found in *Escherichia coli* [32–34] is a site-specific, protein-DNA barrier that has been successfully employed in budding yeast and mammalian cells to induce replication fork stalling [30,31,35]. In yeast, the Tus/*Ter* block creates a polar replication fork barrier that promotes mutagenesis, including sequence deletions and duplications [36], and homologous recombination between inverted repeats [37]. Previous work utilizing the Tus/*Ter* system to study sister chromatid recombination in mammalian cells has shown that bidirectional fork stalling stimulates short tract (STGC) and long-tract gene conversion events (LTGC), as well as the formation of rare, short-span TDs of ~10 kb [22,31,38]. Interestingly, loss of the tumor suppressor/DNA repair factor BRCA1 stimulates the formation of Tus/*Ter*-induced TDs but has no effect on the rare TDs formed in response to an I-SceI endonuclease induced DSB [22]. These findings suggest that stalled forks promote TD formation, and that the mechanism of TD formation at stalled forks has specific genetic requirements that are distinct from those produced through DSB repair.

In this study, we sought to further understand the mechanism of repeat recombination in response to replication fork stalling. We developed a genetic assay in *S. cerevisiae* to determine the frequency of CNVs induced by a site-specific, protein-DNA replication fork barrier. Using this system, we found that a Tus/*Ter*-induced fork block downstream of direct repeats stimulates the formation of TDs and deletions and identify genes that promote or suppress these recombination events.

## Results

### A Tus/*Ter* replication fork block downstream of a direct repeat induces recombination

To evaluate the formation of CNVs, we constructed a direct repeat recombination reporter consisting of two truncated *trp1* fragments, with 426 bp of overlapping homology, that are separated by a *K. lactis URA3* marker (5′Δ-*trp1::URA3::trp1*-3′Δ) (Fig 1A). The regions of homology are 2.09 kb away from each other. The design of this direct repeat reporter is similar to those used previously to detect unequal sister chromatid recombination [39–42]. In contrast to reporters that detect primarily gene conversion between heteroalleles [43–45], our system was designed to detect structural variation. The reporter was integrated in the *S. cerevisiae* genome ~3.8 kb downstream of *ARS607*, a highly efficient origin of replication on chromosome VI, in a haploid strain with the native *trp1-1* locus deleted. In this system, recombination between the two truncated *trp1* alleles can result in 1) the formation of a TD that consists of a full-length *TRP1* gene and two copies

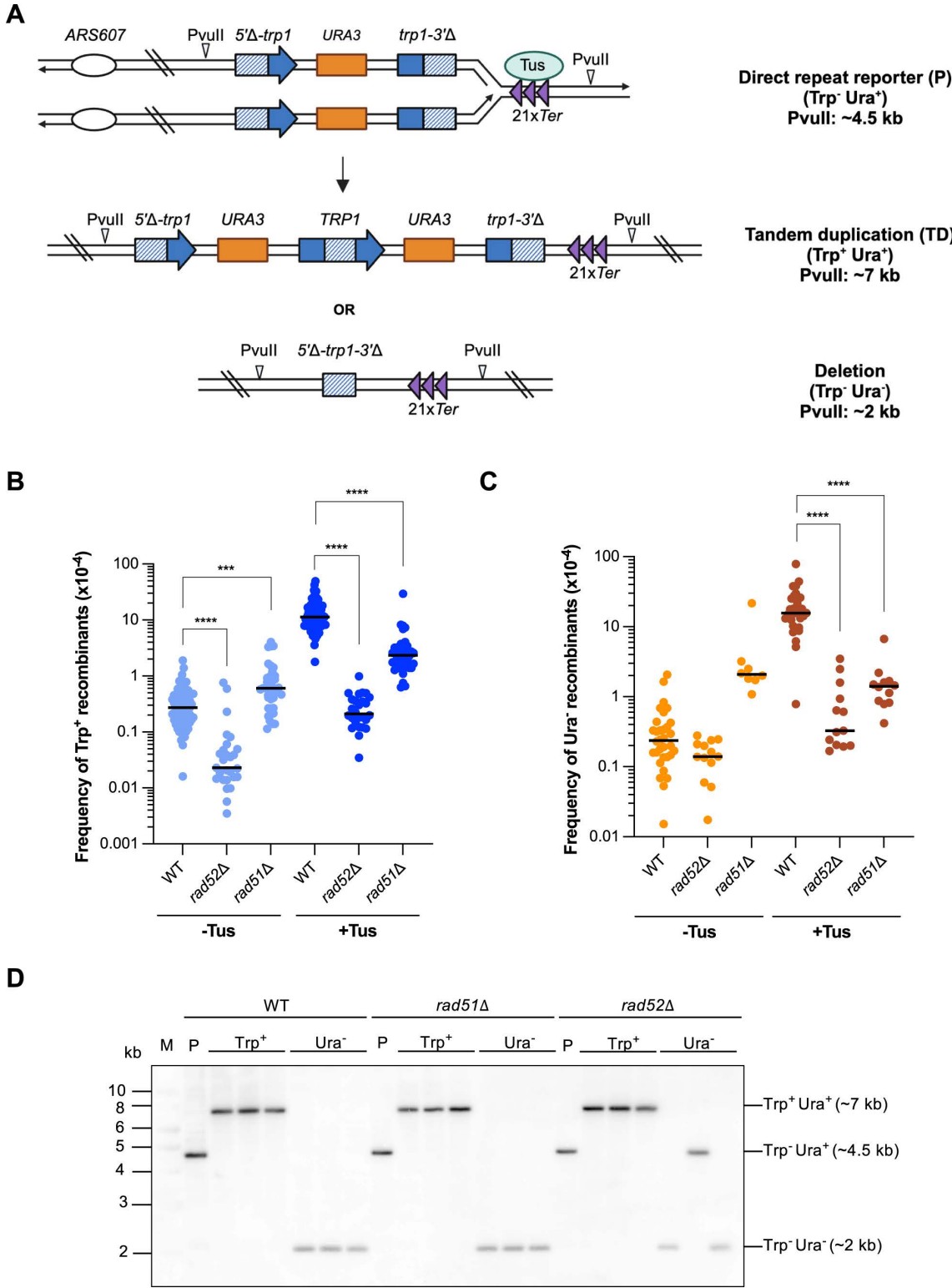

**Fig 1. A Tus/*Ter* block downstream of direct repeats stimulates Rad51 and Rad52-dependent CNV. A.** Schematic of direct repeat reporter consisting of two *trp1* fragments (*5'Δ-trp1* and *trp1-3'Δ*) with 426 base pairs of overlapping homologies (indicated by the shaded lines) that are separated by a *K. lactis URA3* (*URA3*) marker. The reporter is integrated ~3.8 kb distal to *ARS607* on Chr VI with 21x *TerB* repeats integrated downstream of the *trp1*

repeats. Recombination between two truncated *trp1* alleles can yield Trp⁺ recombinants (TDs) or Ura⁻ recombinants (deletions). PvuII restriction sites used for Southern blots in are indicated where appropriate. Created in BioRender. Triplett, M. (2025) https://BioRender.com/loxxdnu. Frequency of Trp⁺ (B) and Ura⁻ (C) recombinants in WT, *rad52Δ*, and *rad51Δ* strains. Statistical significance was determined by one-way ANOVA on log-transformed data with a Bonferroni post-test. p-values are indicated as follows: ns (not significant) $p > 0.05$, *$p < 0.05$, **$p < 0.005$, ***$p < 0.001$, ****$p < 0.0001$. **D.** Southern blot of the non-recombined parental direct repeat reporter strain (P) and the Trp⁺ and Ura⁻ recombination products under conditions in which the Tus/*Ter* block has been induced in WT, *rad51Δ*, and *rad52Δ* strains. "M" indicates the molecular weight size marker (1 kb ladder).

of *URA3* (Trp⁺ Ura⁺), or 2) the deletion of the *URA3* marker (Trp⁻ Ura⁻). These structural changes could arise by unequal sister exchange or long tract gene conversion. Additionally, deletions could be generated by single-strand annealing (SSA) [46]. Hereafter, the products of these recombination events will be referred to as Trp⁺ (TD) and Ura⁻ (deletion) recombinants. Trp⁺ recombinants can be detected by plating cells on synthetic complete medium lacking tryptophan (SC-trp), while Ura⁻ recombinants are detected by plating on medium containing 5-fluoroorotic acid (5-FOA).

To determine the effects of site-specific replication fork stalling on the formation of TDs and deletions, we used a galactose-inducible Tus/*Ter* replication fork barrier to trigger replication fork stalling at a site 138 bp downstream of the *trp1* direct repeats. Specifically, 21 *TerB* repeats in the blocking orientation relative to *ARS607* were integrated downstream of the *trp1-3'Δ* fragment of the recombination reporter (Fig 1A). A $P_{GAL1}$-*HA-Tus* cassette was integrated at the *LEU2* locus. We chose this location for the Tus/*Ter* block with respect to the direct repeat reporter based on previous studies showing that a Tus/*Ter* barrier integrated ~5 kb telomeric to *ARS607* was able to stimulate mutagenesis or recombination of sequences origin proximal to the *Ter* repeats in response to fork stalling [36,37].

We found that the spontaneous frequency of Trp⁺ recombinants (0.27 x 10⁻⁴, rate of 3.94 x 10⁻⁶/cell/generation) in the absence of Tus expression was comparable to those determined in previous studies using similarly-constructed genetic reporters [39,40,42]. Upon expression of Tus, we observed a ~42-fold induction of Trp⁺ recombinants compared to spontaneous frequencies (0.27 x 10⁻⁴ to 11.3 x 10⁻⁴) (Fig 1B, S1 Table) and a ~65-fold induction of Ura⁻ events compared to spontaneous frequencies (0.24 x 10⁻⁴ to 15.6 x 10⁻⁴) (Fig 1C, S2 Table). These data demonstrate that a Tus/*Ter* block downstream of a direct repeat reporter stimulates recombination leading to the formation of either TDs or deletions.

## Tus/*Ter*-induced recombination is dependent on both Rad52 and Rad51

Two of the key proteins involved in homologous recombination in yeast are the Rad51 recombinase and the Rad52 mediator protein [46]. Rad51 conducts the search for homology and catalyzes strand exchange, while Rad52 mediates the loading of Rad51 onto RPA-coated ssDNA. In addition to Rad51 loading, Rad52 catalyzes annealing of single-stranded DNA [47,48]. We previously demonstrated that HR induced by a Tus/*Ter* block downstream of inverted repeats is dependent on Rad51 strand invasion and Rad52 strand annealing activities [37]. Additionally, gene conversion between direct repeats induced by the *RTS1* replication-fork barrier in *S. pombe* requires Rad51 and Rad52 [29].

To confirm the importance of HR in the formation of Trp⁺ and Ura⁻ events induced by a Tus/*Ter* barrier, we measured the frequency of recombination in strains with either *RAD52* or *RAD51* deleted. Loss of Rad52 reduced spontaneous Trp⁺ frequencies by 13–14-fold (Fig 1B), which aligns with the findings of a previous study using a similar direct repeat recombination reporter [39]. Also consistent with this previous study, we observed that spontaneous Trp⁺ frequencies were higher for *rad51Δ* than for the wild-type (WT) strain. While the spontaneous Ura⁻ frequencies of the *rad52Δ* and *rad51Δ* mutants were not statistically different from the WT strain, the overall trends for these mutants with respect to WT were consistent with Trp⁺ frequencies (Fig 1C). In the presence of a replication fork block, we found that loss of Rad52 reduced the frequency of Trp⁺ and Ura⁻ recombination events by approximately 50-fold, and loss of Rad51 led to a ~5-fold decrease in the frequency of Trp⁺ recombinants (11.3 x 10⁻⁴ to 2.3 x 10⁻⁴) and a ~11-fold decrease in the frequency of Ura⁻ recombinants (15.6 x 10⁻⁴ to 1.4 x 10⁻⁴) (Fig 1B, 1C and S1 and S2 Tables). It should also be noted that the overall induction of recombination in the presence vs. the absence of Tus in the *rad51Δ* mutant (~4-fold increase for Trp⁺ and no

induction for Ura⁻ events) was much lower than for the WT strain (~42-fold increase for Trp⁺ and ~65-fold increase for Ura⁻ events). Consistent with other studies using the Tus/*Ter* system in yeast, we did not observe a growth defect for WT or HR-deficient mutants when Tus is expressed (S1 Fig).

To verify the expected structure of Trp⁺ and Ura⁻ recombinants, we isolated genomic DNA from independent recombinants for *Pvu*II digestion and Southern blot analysis. As a control, we analyzed colonies from the parental (P), non-recombined direct repeat reporter strain of each genotype, confirming a *Pvu*II fragment of the expected size. We observed that all Trp⁺ recombinants analyzed from WT, *rad52Δ* and *rad51Δ* mutant strains do indeed have the predicted ~7 kb fragment size (Fig 1A, 1D). All Ura⁻ recombinants analyzed from the WT and *rad51Δ* backgrounds contained a fragment that was the predicted size of the deletion product (~2 kb), while one out of the three Ura⁻ colonies in the *rad52Δ* background contained a fragment the length of the direct repeat reporter. This event is most likely due to a point mutation within the *URA3* gene since the *rad52Δ* mutant is known to have elevated spontaneous mutagenesis [49–51]

Previous studies have shown that Rad59, a paralog of Rad52, promotes recombination between inverted repeats in the absence of Rad51 [37,52]. Although we found that loss of Rad59 alone had no significant impact on spontaneous or Tus-induced recombination frequencies, the *rad51Δ rad59Δ* double mutant displayed a synergistic decrease in Tus/*Ter*-induced Trp⁺ events (S2 Fig and S1 and S2 Tables). However, the frequency of Ura⁻ events was not significantly different for the *rad51Δ rad59Δ* double mutant compared to the *rad51Δ* single mutant. These data confirm that canonical HR proteins are important for replication fork restart in the event of fork stalling induced by a Tus/*Ter* barrier. Hereafter, data for Trp⁺ recombination frequencies are presented in the figures and Ura⁻ data are only shown when a mutation significantly impacts one type of event more than the other. The Trp⁺ and Ura⁻ recombination frequencies for all strains tested are presented in S1 and S2 Tables.

## Loss of the MRX complex increases the frequency of Tus/*Ter*-induced TDs

Previous studies have shown that end resection is required for replication fork restart at the *RTS1* barrier in *S. pombe* and in the formation of Tus/*Ter*-induced recombination between inverted repeats [37,53]. Therefore, we anticipated that end resection would be required to generate Tus/*Ter*-induced CNVs. The Mre11 nuclease functions as a part of the Mre11-Rad50-Xrs2 (MRX) complex to initiate the degradation of 5′ strands to create 3′ ssDNA tails (Fig 2A) [54]. Surprisingly, we observed that although *mre11Δ* and *xrs2Δ* mutants had spontaneous Trp⁺ frequencies comparable to the WT strain, both mutants exhibited a significant increase (~4–5-fold, $11.3 \times 10^{-4}$ to $42.3 \times 10^{-4}$ or $50.6 \times 10^{-4}$, respectively) in the frequency of Tus-induced recombination events (Fig 2B and S1 and S2 Tables). We found that the increased frequency of Tus-induced Trp⁺ recombinants in the *mre11Δ* mutant was dependent on Rad51, indicating that they form by a strand invasion mechanism (Fig 2C). These data suggest that the MRX complex may suppress the formation of TDs in the presence of a replication fork block. Loss of Sae2, which is required to activate the Mre11 nuclease [55], or inactivation of Mre11 nuclease (*mre11-H125N* mutation), did not alter the frequency of Tus-induced recombination (Fig 2B, S1 Table), suggesting that the *mre11Δ* hyper-recombination phenotype is independent of its function in resection initiation. It is important to note that Mre11 nuclease is not essential to initiate resection of ends lacking covalent modifications in budding yeast, and resection can initiate directly by the long-range resection factors, Exo1 and Sgs1-Dna2 [54].

The MRX complex has several known and proposed roles beyond initiating short-range resection, including non-homologous end joining (NHEJ) and checkpoint activation through the Tel1 kinase [56] (Fig 2A). To assess the potential role of these MRX-related processes in the formation of Tus/*Ter*-induced TDs, we measured the recombination frequencies of strains lacking Dnl4 and Tel1 (Fig 2B). However, these strains exhibited approximately the same Tus-induced Trp⁺ frequencies as in WT cells, so it is unlikely that perturbing MRX's role in these processes contributes to the observed hyper-recombination phenotype of the MRX mutants.

A previous study found that the association of various replisome components with replication forks is disrupted in MRX mutants treated with hydroxyurea (HU), suggesting that MRX is important for replisome stability at stalled forks [57]. We

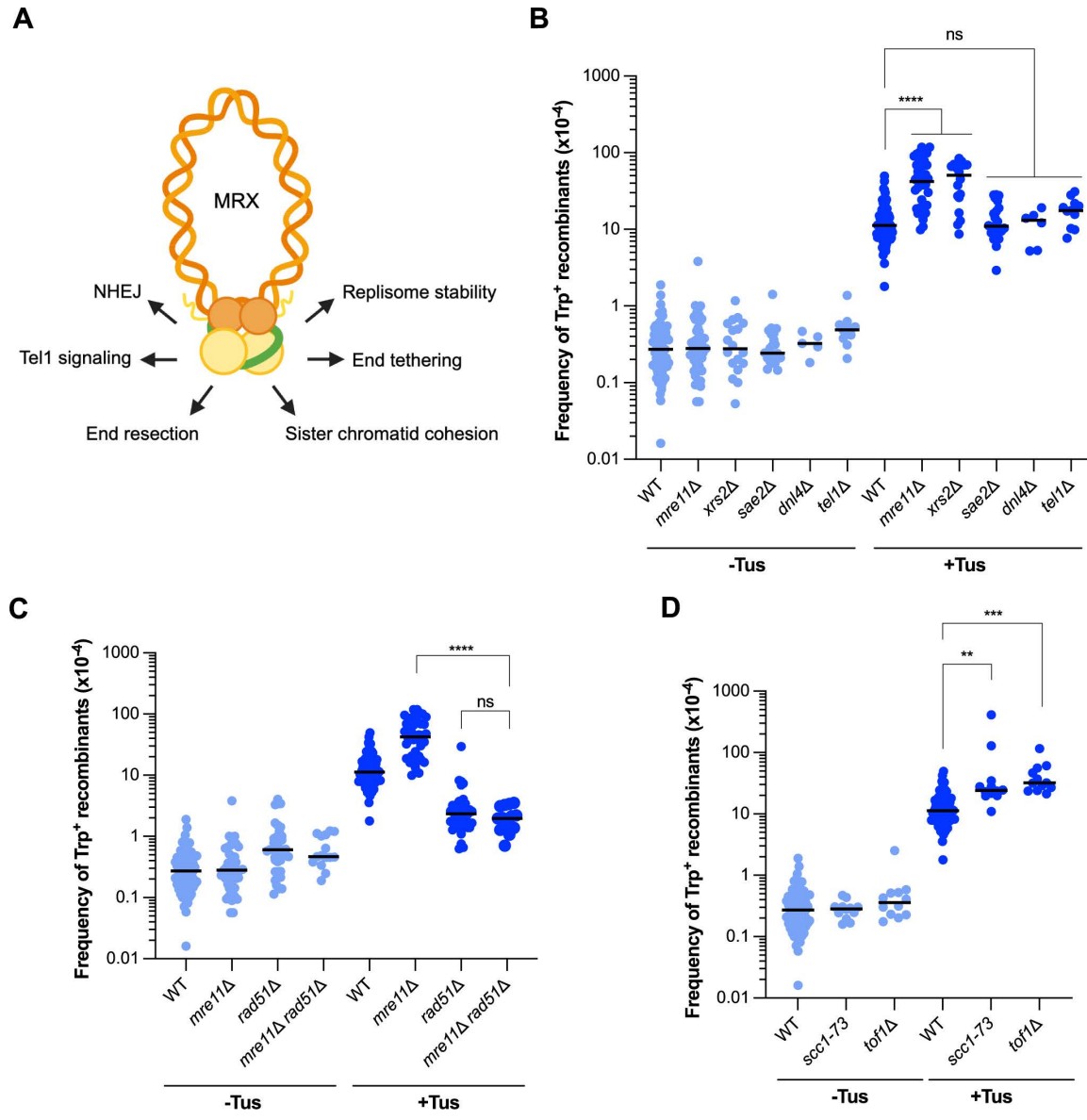

**Fig 2. Loss of MRX components and sister chromatid cohesion increases the frequency of Tus/*Ter*-induced TDs.** A. The MRX complex and its functions. Key: Mre11-yellow, Rad50-orange, Xrs2-green. Created in BioRender. Triplett, M. (2025) https://BioRender.com/j1vpn7o. B. Frequencies of Trp⁺ recombinants in WT and *mre11Δ*, *xrs2Δ*, *sae2Δ*, *dnl4Δ*, and *tel1Δ* strains. C. Frequencies of Trp⁺ recombinants in WT, *mre11Δ*, *rad51Δ* and *mre11Δ rad51Δ* strains. D. Frequencies of Trp⁺ recombinants in WT and *scc1-73* and *tof1Δ* strains. Statistical significance for recombination assays was determined by one-way ANOVA on log-transformed data with a Bonferroni post-test. p-values are indicated as follows: ns (not significant) $p > 0.05$, *$p < 0.05$, **$p < 0.005$, ***$p < 0.001$, ****$p < 0.0001$.

reasoned that decreased replisome stability at the Tus/*Ter* block in the absence of MRX might impede direct replication restart and thus contribute to the observed hyper-recombination phenotype. To test this idea, we performed ChIP-qPCR to detect the presence of the Mcm2–7 helicase upstream of the block. WT and *mre11Δ* cells were arrested in G1 with α-factor and then released into S phase in the presence or absence of galactose (Tus expression). Cells were collected at 0-, 35-, 45-, and 55-minutes post-release into S phase (S3A Fig). These time points were chosen based on previous

studies showing that the Tus/*Ter* block is the strongest 35 minutes after release into S-phase [30,36]. ChIP was performed with Mcm2–7 polyclonal antibodies [58,59], and qPCR was then performed using primers to amplify the 138 bp region between the *trp1–3′Δ* end of the direct repeat reporter and the first *TerB* site. While we did observe an accumulation of Mcm2–7 at the site upstream of the block in cells where Tus had been induced compared to non-induced conditions (an indication that the replisome was indeed stalled in Tus-induced cells) 35–45 minutes after release from G1 we did not observe any notable difference in Mcm2–7 levels between WT and *mre11Δ* cells (S3B Fig). Discrepancies between our results and Tittel-Elmer et al. [57] could be due to the difference in the mechanism of fork stalling between Tus/*Ter*, a transient, protein-DNA barrier, and HU, which slows forks through depletion of dNTPs. It is possible that global fork stalling by HU in the *mre11Δ* background produces a stronger defect in replisome stability than what can be observed at a single fork stalling site.

### Impaired sister chromatid cohesion increases the frequency of TDs

The Rad50 component of the MRX complex is a member of the structural maintenance of chromosomes (SMC) family of proteins, whose main structural features include ATPase motifs in their N-terminal and C-terminal domains separated by two coiled-coil domains, and a hinge domain [56,60]. Evidence has suggested that the Rad50 subunit of MRX is important for the tethering of DSB ends, as well as sister chromatid tethering at DSBs [61–66]. MRX has also been shown to be involved in chromatin remodeling at HU-stalled replication forks, which both facilitates resection and promotes the recruitment of cohesin [67]. We reasoned that if loss of MRX results in a reduction in sister chromatid cohesion, it is possible that other mutants defective in cohesion could also have a similar hyper-recombination phenotype. We evaluated the role of sister chromatid cohesion in the formation of Tus-induced TDs by measuring the recombination frequencies in a strain containing a cohesin complex conditional mutation (*scc1-73)* and in a strain lacking Tof1, a component of the cohesion establishment/fork protection factor Tof1-Csm3 (Fig 2D). Interestingly, these sister chromatid cohesion-related mutants did indeed exhibit a hyper-recombination phenotype for Tus-induced TDs, similarly to that of the MRX mutants, suggesting that loss of sister chromatid cohesion may promote TD formation upon fork stalling.

### Long-range resection by Exo1 is important for Tus/*Ter*-induced CNVs

The generation of 3′ ssDNA ends through short-range resection by MRX allows for the recruitment of Exo1 and Sgs1-Dna2 to promote long-range resection [54]. To assess whether the hyper-recombination phenotype of MRX mutants could be a result of either impaired recruitment of long-range resection factors or a decrease in resection overall, we measured recombination frequencies of *exo1Δ* and *sgs1Δ* mutant strains. Consistent with other studies [37,53], loss of Exo1 resulted in a ~3-fold reduction of the frequency of Tus/*Ter*-induced Trp+ recombinants (11.3 x 10^-4 to 3.4 x 10^-4) (Fig 3A, S1 Table). Interestingly, the *exo1Δ* mutation caused a ~43-fold decrease in the frequency of deletions suggesting that these events are more dependent on extensive end resection than TDs (Fig 3B, S2 Table). Loss of Sgs1 resulted in an increase in both spontaneous and Tus/*Ter*-induced recombination (Fig 3A and 3B) consistent with a hyper-recombination phenotype previously reported [36,68]. However, it is likely that the hyper-recombination phenotype of *sgs1Δ* is a result of the disruption of its role in the dissolution of recombination intermediates [69,70] and not its role in long-range resection. Loss of Exo1 in the *sgs1Δ* background reduced the frequency of Tus-induced TDs by 11-fold, and deletions by 950-fold compared with the *sgs1Δ* single mutant (Fig 3A and 3B). Thus, long-range end resection is important for Tus-induced recombination, particularly for deletion events, and is mostly due to Exo1 activity.

Since previous studies have shown that over-expression of Exo1 can suppress the DNA damage sensitivity of *mre11Δ* and *rad50Δ* mutants [71–73], we tested whether it could suppress the hyper-recombination phenotype of the *mre11Δ* mutant. Overexpression of Exo1 reduced the frequency of Tus-induced Trp+ recombinants in the *mre11Δ* background to that of WT, while having no noticeable impact on the frequency of these events in the WT background (Fig 3C). These results suggest that a resection defect may contribute to the hyper-recombination phenotype displayed by the MRX mutants.

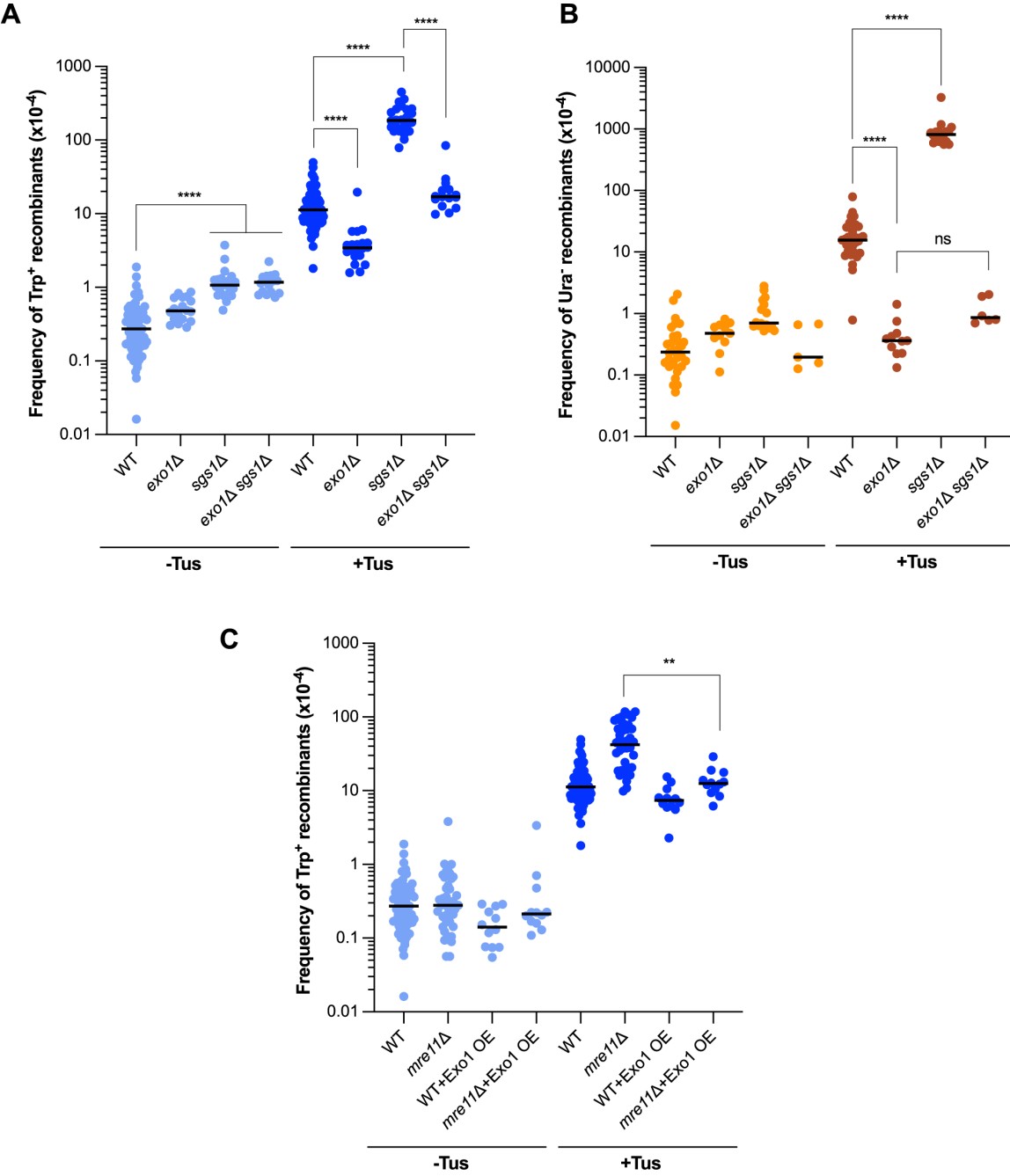

**Fig 3. Long-range resection by Exo1 is important for Tus/*Ter*-induced TD formation.** A. Frequencies of Trp⁺ recombinants in WT, *exo1Δ*, *sgs1Δ*, and *exo1Δ sgs1Δ* strains. B. Frequencies of Ura⁻ recombinants in WT, *exo1Δ*, *sgs1Δ*, and *exo1Δ sgs1Δ* strains. C. Frequencies of Trp⁺ recombinants in WT and *mre11Δ* strains in which Exo1 is overexpressed compared to strains without Exo1 overexpression. Statistical significance for recombination assays was determined by one-way ANOVA on log-transformed data with a Bonferroni post-test. p-values are indicated as follows: ns (not significant) p>0.05, *p<0.05, **p<0.005, ***p<0.001, ****p<0.0001.

## The formation of Tus/*Ter*-induced TDs requires the Mph1 translocase and structure-selective endonucleases

We next measured the frequency of recombination in strains lacking replication fork remodeling proteins or structure-selective endonucleases (SSEs) (Fig 4). The frequencies of spontaneous TD formation were not altered by the loss of Rad5 or Mph1 alone or the by the loss of both factors (Fig 4A). However, loss of Mph1 led to a ~ 4-fold and ~9-fold decrease in the frequencies of Tus-induced Trp⁺ and Ura⁻ recombinants, respectively (S1 and S2 Tables). Loss of Rad5 alone did not have a significant effect on the frequency of Tus-induced TDs and did not further decrease the recombination frequency of the *mph1Δ* mutant. These results suggest the possibility that fork reversal mediated by Mph1 is necessary to restart stalled forks generated by a Tus/*Ter* block. However, it is important to note that Mph1 also plays a role in D-loop dissociation, and this function could contribute to the formation of recombinants in response to fork stalling [74].

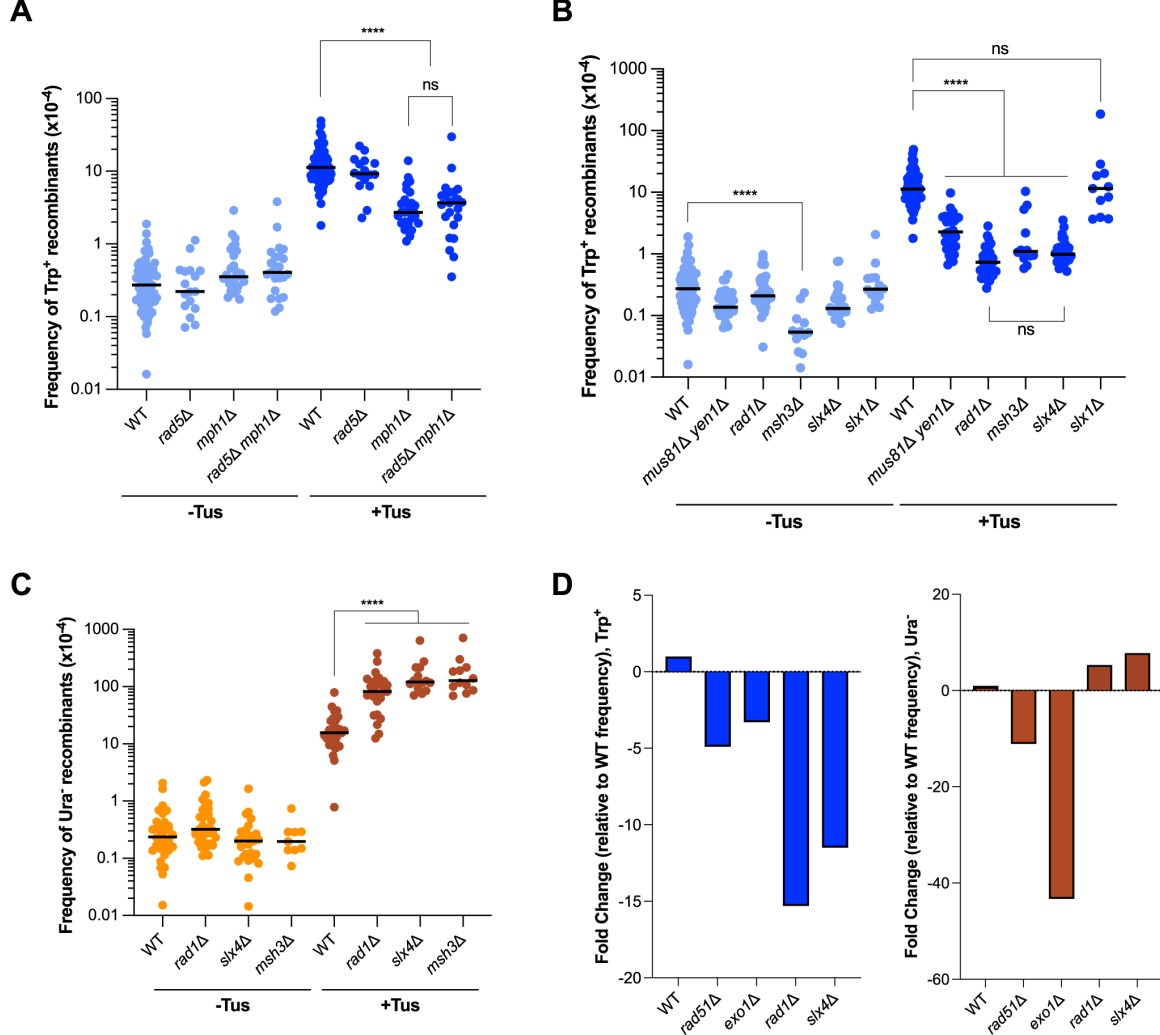

**Fig 4. Tus-induced TDs and deletions have specific genetic requirements. A.** Frequencies of Trp⁺ recombinants in WT and strains lacking genes involved in replication fork reversal and regression (*rad5Δ, mph1Δ,* and *rad5Δ mph1Δ*). **B.** Frequencies of Trp⁺ recombinants in WT and strains lacking SSEs (*mus81Δ yen1Δ, rad1Δ, slx1Δ, slx4Δ*). **C.** Frequencies of Ura⁻ recombinants in WT and strains lacking Rad1, Slx4 or Msh3. Statistical significance was determined by one-way ANOVA on log-transformed data with a Bonferroni post-test. p-values are indicated as follows: ns (not significant) p > 0.05, *p < 0.05, **p < 0.005, ***p < 0.001, ****p < 0.0001. **D.** Fold change in the Tus/Ter-induced recombination frequency relative to WT for *rad51Δ, exo1Δ, rad1Δ* and *slx4Δ* mutants.

We also assessed the involvement of the scaffold protein, Slx4, and its endonuclease binding partners (Slx1 and Rad1-Rad10) in the formation of TDs (Fig 4B) [75–78]. Loss of Slx4 significantly reduced the frequency of Tus/*Ter*-induced Trp⁺ recombinants, suggesting that SSEs are important for the formation of duplications in response to fork stalling. Loss of Slx1 had no effect on the frequency of Tus-induced recombinants. The Rad1-Rad10 nuclease cleaves branched DNA structures at the transition between dsDNA and ssDNA, and has previously been shown to be important to remove 3′-heterologous flaps formed during strand invasion and SSA, as well as large loop heteroduplexes [79–81]. Like Slx4, loss of Rad1 or Rad10 significantly reduced the frequency of Tus/*Ter*-induced TDs almost to spontaneous levels (Fig 4B, S1 Table), suggesting that the removal of 3′ heterologous flaps or loops by Rad1-Rad10 is required for replication-associated recombination events leading to the formation of TDs. There was no further decrease in the frequency of Tus/*Ter*-induced TDs in the *rad1Δ slx4Δ* double mutant indicating that these factors function together to promote recombination (S4 Fig). These results complement a study in mouse cells showing that SLX4-XPF regulates HR at replication forks stalled by the Tus/*Ter* barrier [82]. The Msh2-Msh3 complex functions with Rad1-Rad10 in 3′-heterologous flap removal and in the repair of large loop heteroduplexes during meiotic recombination [81,83]. Similarly to Rad1, loss of Msh3 reduced the frequency of Tus/*Ter*-induced TDs (Fig 4B). Unexpectedly, the frequency of Tus/*Ter*-induced deletions increased in the absence of Slx4, Rad1 or Msh3 (Fig 4C, S2 Table). This finding indicates that the Rad1-Rad10 nuclease inhibits deletion formation, which contrasts with its role in SSA where it promotes deletions [84]. While *slx4Δ* and *rad1Δ* mutations conferred opposite effects on the formation of TDs and deletions, the *rad51Δ* and *exo1Δ* mutations both resulted in a stronger defect in deletion formation than TDs (Fig 4D).

Mus81-Mms4 is the main nuclease responsible for cleaving replication and recombination intermediates, with Yen1 providing a backup function [85–87]. While *mus81Δ* and *yen1Δ* single mutations had no effect on the frequency of Tus-induced recombination, the *mus81Δ yen1Δ* double mutant exhibited a ∼5-fold decrease in the frequency of Tus-induced TDs (Fig 4B, S1 Table). This finding suggests that cleavage of a reversed replication fork or a recombination intermediate is important for the formation of TDs in response to fork stalling. In *S. cerevisiae*, Slx4 does not interact directly with Mus81-Mms4, but can interact indirectly via Dpb11 in G2-phase cells [88,89]. The observation that *slx4Δ* and *rad1Δ* mutations confer a stronger defect in the formation of Tus/*Ter*-induced TDs than *mus81Δ*, and are epistatic, suggests that Slx4 functions primarily through its role with Rad1-Rad10.

## Loss of Rad52 or Rad51 further decreases Tus/*Ter*-induced TD formation in the *rad1Δ* background

Since the *rad1Δ* mutation caused a profound defect in the generation of Tus/*Ter*-induced TDs, we next sought to explore the relationship between Rad1 and HR proteins (Fig 5). Similarly to what has been reported in previous studies, we found that both the *rad1Δ rad52Δ* and *rad1Δ rad51Δ* double mutants exhibited a synergistic decrease in spontaneous Trp⁺ recombination frequencies [39,42,44]. In our system, spontaneous Trp⁺ events are below the detection limit in the *rad1Δ rad52Δ* double mutant. Although we observed some induction of Trp⁺ recombinants in the presence of a Tus/*Ter* block, the frequency was ∼100-fold lower than that of the *rad1Δ* and *rad52Δ* single mutants (Fig 5A). Similarly, the frequency of Tus-induced Trp⁺ events in the *rad1Δ rad51Δ* double mutant was reduced by 100- and 15-fold compared with the *rad51Δ* and *rad1Δ* single mutants, respectively (Fig 5B). The observation that both *rad1Δ rad52Δ* and *rad1Δ rad51Δ* double mutants exhibit lower frequencies of Tus-induced TDs than their respective single mutants suggests that Rad1 can function independently of HR to promote TD formation, perhaps by acting at sites of polymerase-mediated replication slippage. Simultaneous loss of HR factors and Rad1 had less of an impact on deletion formation than TDs (Fig 5C and 5D). Notably, the hyper-deletion phenotype of the *rad1Δ* mutant is Rad51 and Rad52 dependent, consistent with deletions arising by a strand invasion mechanism.

## Cas9 nickase-induced fork breakage bypasses the need for Mph1 and HJ resolvases in formation of TDs

We reasoned that if the role of Mph1 and SSEs is to reverse and cleave the stalled replication fork, respectively, then using Cas9 nickase to generate a replication-dependent DSB would bypass the requirement for these factors

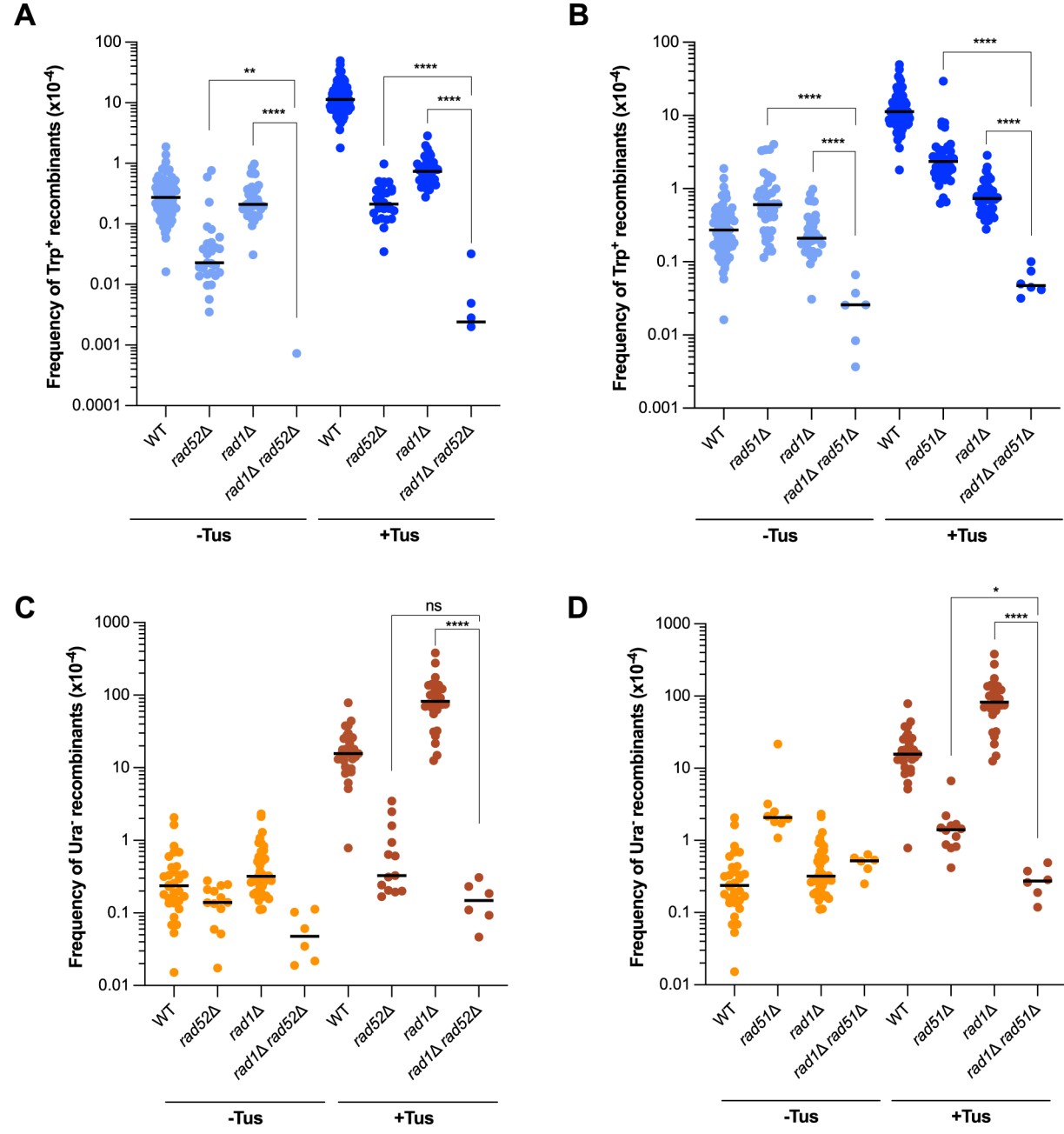

**Fig 5. Loss of Rad52 or Rad51 further reduces TDs in the *rad1*Δ background.** Frequencies of Trp+ (A) and Ura- (C) recombinants in WT, *rad1*Δ, *rad52*Δ and the *rad1*Δ *rad52*Δ double mutant. Frequencies of Trp+ (B) and Ura- (D) recombinants in WT, *rad1*Δ, *rad51*Δ and the *rad1*Δ *rad51*Δ double mutant. Statistical significance was determined by one-way ANOVA on log-transformed data with a Bonferroni post-test. p-values are indicated as follows: ns (not significant) p > 0.05, *p < 0.05, **p < 0.005, ***p < 0.001, ****p < 0.0001.

(Fig 6A). We modified a previously described system for inducible expression of Cas9$^{D10A}$, hereafter referred to as nCas9, in strains with a constitutively expressed gRNA that targets the leading-strand template of a sequence 138 bp downstream of the *trp1-3'*Δ fragment (gRNA6 that was used to integrate the 21 *TerB* repeats via Cas9-mediated genome editing) [90,91]. An estrogen receptor domain was fused to nCas9 so that both expression and nuclear

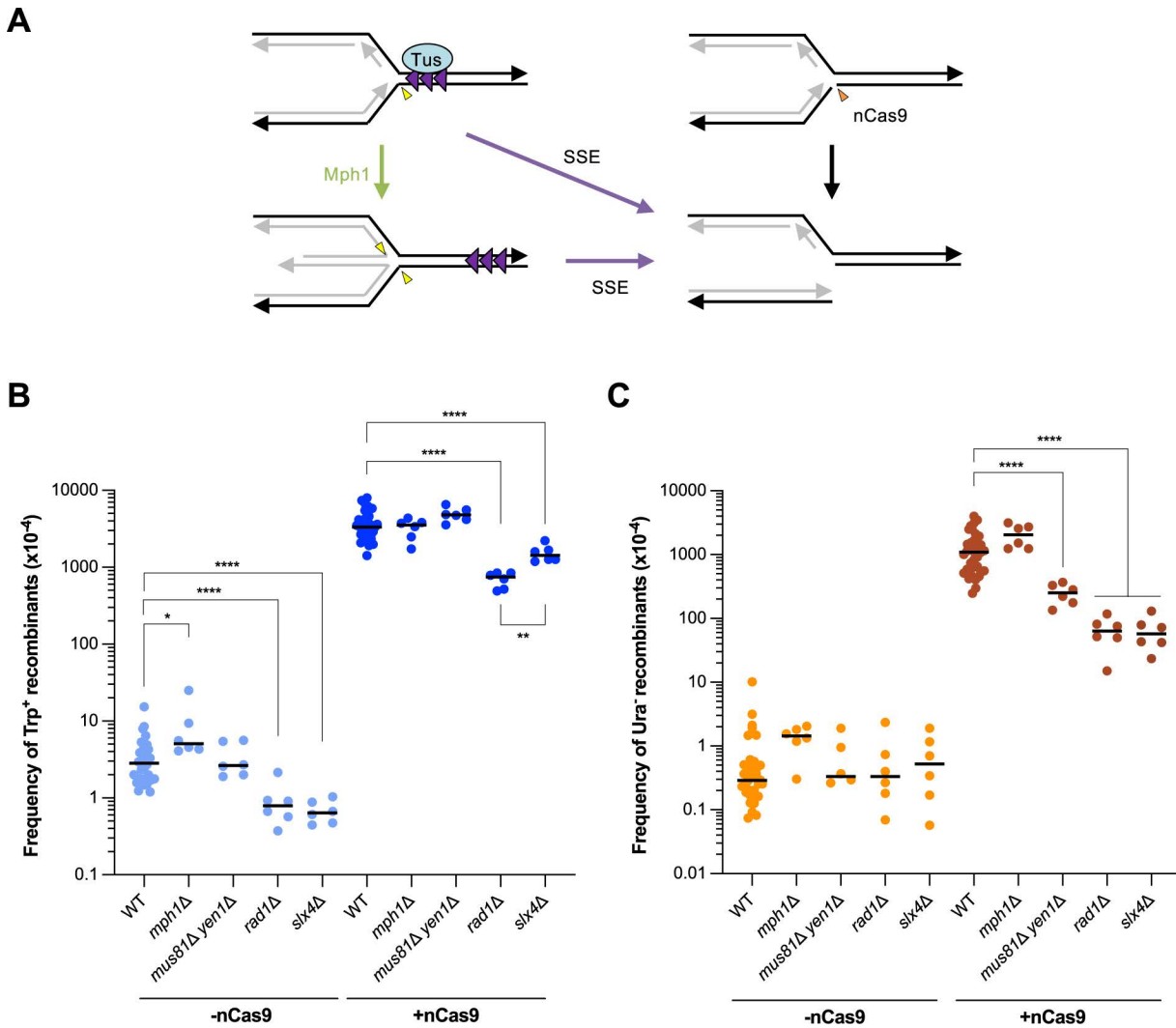

**Fig 6. Cas9$^{D10A}$-induced fork breakage bypasses the need for Mph1 and HJ resolvases in formation of TDs. A.** Schematic of a replication fork stalled at the Tus/Ter barrier and fork collapse induced by a nick on the leading strand template. Frequencies of Trp$^+$ (B) and Ura$^-$ (C) recombinants under non-induced (no β-estradiol) and induced (+β-estradiol) conditions in WT, *mph1Δ*, *mus81Δ yen1Δ*, and *rad1Δ* strains expressing Cas9$^{D10A}$ and gRNA6. Statistical significance was determined by one-way ANOVA on log-transformed data with a Bonferroni post-test. p-values are indicated as follows: ns (not significant) $p > 0.05$, *$p < 0.05$, **$p < 0.005$, ***$p < 0.001$, ****$p < 0.0001$.

localization of the nickase depend on the addition of β-estradiol, in a similar manner to that described previously [90]. We found that a nick induced downstream of the direct repeat reporter increased Trp$^+$ and Ura$^-$ recombination frequencies to 3329 X 10$^{-4}$ and 1095 x 10$^{-4}$, respectively (Fig 6B, 6C and S3 Table). The large increase in TDs is consistent with recent studies showing expansion of repeat arrays by nCas9-induced nicks [92,93]. The Trp$^+$ frequency under non-inducing conditions was 10-fold higher than the true spontaneous frequency, indicating leaky expression of nCas9 (S5A Fig). Although a nick on the leading-strand template is thought to convert to a one-ended DSB by CMG run-off at the nick [94], studies in yeast have shown that replication-dependent two-ended DSBs can result from leading-strand template nicks [91,95]. It is unclear whether the recombination events induced by nCas9 in this context result from repair of a one- or two-ended DSB.

Although the *mph1Δ* and *rad1Δ* mutants were not sensitive to nick induction, growth of the *mus81Δ yen1Δ* strain was reduced with nCas9 expression (S5B Fig). Expression of nCas9-gRNA6 in the *mph1Δ* and *mus81Δ yen1Δ* mutants increased the frequency of Trp⁺ recombinants to the same level as in WT cells (Fig 6B, S3 Table). These data are consistent with the idea that generation of Tus/*Ter*-induced TDs requires fork reversal and cleavage of branched DNA structures, and these roles can be bypassed by fork breakage. The *rad1Δ* and *slx4Δ* mutants showed 4.5 and 2.3-fold decreases, respectively, in the frequency of nCas9-induced Trp⁺ recombinants compared to WT. Furthermore, in contrast to Tus/*Ter*-induced deletions, we found that nCas9-induced deletions are Rad1 and Slx4 dependent (Fig 6C). The requirement for Rad1 and Slx4 to generate both classes of recombinants induced by a replication-associated DSB is consistent with the role of Rad1-Rad10 in cleavage of heterologous flaps that form during Rad51-dependent strand invasion. However, the opposite phenotype of the *rad1Δ* and *slx4Δ* mutants with regards to deletion formation in response to different types of replication stress suggests a unique role for Slx4-Rad1-Rad10 during recombination at a protein-induced fork barrier.

## Discussion

We have established a direct repeat recombination reporter capable of detecting either TDs or deletions. We found that a Tus/*Ter* block downstream of direct repeats stimulates both classes of events by a mechanism that involves Rad51-catalyzed strand invasion, long-range DNA end resection, and fork reversal/regression. While most of the mutants tested in this study showed a similar fold change in the frequency of TDs and deletions, the *rad1Δ*, *rad10Δ* and *slx4Δ* mutants were the exception. Surprisingly, we found that the Slx4 scaffold protein and its nuclease binding partner, Rad1-Rad10 [75,77,78], are required for the generation of TDs but not deletions. Indeed, the frequency of deletions increased in the absence of Slx4, Rad1 or Rad10, indicating that SSA does not contribute to the formation of deletions in this context. The finding that Tus/*Ter*-induced deletions are Rad51-dependent is also consistent with the proposal that deletions occur by a strand invasion mechanism rather than by SSA [47]. These observations suggest the possibility of an intermediate for both classes of event that is cleaved by Rad1-Rad10 in a manner that generates TDs and eliminates deletions.

The partial requirement for the Mph1 translocase in the generation of Tus/*Ter*-induced duplications and deletions suggests that both products involve replication fork reversal (Fig 7). This idea is further supported by the observation that fork collapse by nCas9, which would produce a recombinogenic end independent of fork remodeling, bypasses the requirement for Mph1 in formation of recombinants. Following fork reversal, the paired nascent strands would need to be resected to create a 3′ ssDNA tail containing one of the repeated sequences for Rad51-dependent strand invasion of the annealed parental strands. Fork reversal would place the 3′ end of the *trp1-3′Δ* allele at the invading end. If invasion of the *5′-trp1Δ* allele occurred, DNA synthesis that continued to the end of the repeat would be sufficient to generate a full-length *TRP1* gene. There are multiple ways in which this recombination intermediate could be resolved. One possible mechanism would involve convergence of a replication fork initiated from a downstream origin with the D-loop intermediate (Fig 7). Since the Tus/*Ter* block is polar in yeast [30], replication from a downstream origin would not be prevented by the *Ter* repeats. The extended invading strand could then ligate to the lagging strand at the converging fork resulting in the formation of a heterologous loop comprised of the nascent strand of the reversed fork up to the *trp1-3′Δ* allele. A study of meiotic recombination involving large hemizygous insertions showed that Rad1-Rad10 is required for repair of a heteroduplex intermediate containing a large loop in a manner that duplicates the insertion to generate a gene conversion product [81]. We suggest that Rad1-Rad10 acts in a similar fashion to cleave the strand opposite the large loop heterology. Fill-in DNA synthesis at the resulting gap would result in a TD. A requirement for Rad1-Rad10 to remove the heterology from the invading end prior to initiation of repair synthesis could also contribute the decreased frequency of TDs but would not explain the increased frequency of deletions in the *rad1Δ* mutant. If fork stalling and reversal occurred >140 bp upstream of the *Ter* repeats, then the 3′ invading end would be homologous to sequences within the reporter, obviating the need for non-homologous tail trimming by Rad1-Rad10.

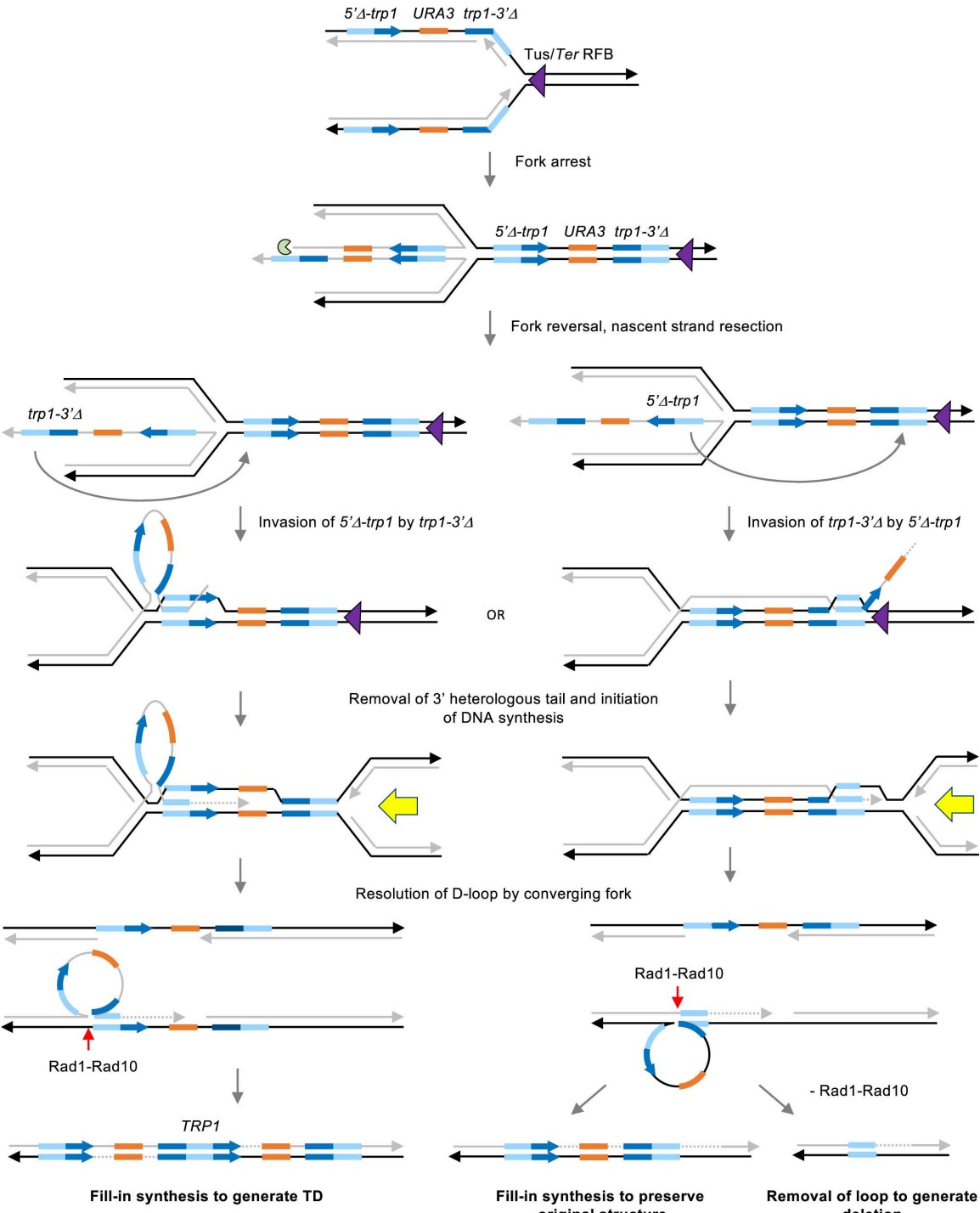

**Fig 7. Model for CNV formation at a stalled replication fork.** Upon encountering a Tus/*Ter* barrier, the replication fork could undergo reversal medi-ated by Mph1. The reversed fork could be resected through the nuclease activity of Exo1 to form a 3' ssDNA overhang that includes the 3'-truncated repeat. The end of the reversed nascent strand could invade the annealed parental strands at the 5'-truncated repeat, mediated by Rad52 and Rad51. Following DNA synthesis to copy the 5' end of *TRP1*, the D-loop could be resolved by an incoming replication fork resulting in the formation of a large heterologous loop. Cleavage of the strand opposite the loop heterology by Rad1-Rad10, followed by fill-in synthesis would generate a full-length copy of *TRP1* and duplicate most of the reporter. A deletion could occur if invasion initiated from the 5' truncated repeat, copying sequence downstream of

PLOS Genetics

the 3′ truncated repeat. Resolution by an incoming replication fork would loop out the parental strand. Cleavage of the nascent strand opposite the loop heterology by Rad1-Rad10, followed by fill-in synthesis would preserve the original structure of the reporter. A deletion product would require cleavage of the ssDNA loop, or segregation of the strands at the next division cycle. The region of homology shared by the repeats is shown in mid-blue, while the 3′ end and 5′ end of *TRP1* are shown in dark blue. Parental strand is indicated by black lines, nascent strand by gray lines, and repair-associated DNA synthesis by a dashed gray line.

Strand invasion of the *trp1-3′Δ* allele on the annealed parental strands by the *5′-trp1Δ* allele of the reversed fork would have the potential to form a deletion by a similar mechanism (Fig 7). A deletion event would require more extensive resection of the 5′-terminated strand of the reversed fork to expose the *5′-trp1Δ* allele, explaining the greater dependence of deletions than duplications on Exo1. The dsDNA loop out adjacent to the D-loop intermediate could be unwound by the replicative helicase resulting in formation of a large heterologous loop on one sister chromatid. In this case, cleavage of the strand opposite the heterologous loop by Rad1-Rad10, followed by fill-in DNA synthesis, would preserve the original structure of the direct repeat. A deletion could only be generated if the unpaired large loop was cleaved and degraded instead of the strand opposite the loop, or by segregation of the unrepaired loop in the next cell cycle. Thus, we propose that the opposite effect of the *rad1Δ* and *slx4Δ* mutations on the generation of TDs and deletions is due to biased cleavage of a large loop heterology by Rad1-Rad10. How the strand opposite the loop could be targeted by Rad1-Rad10 is not entirely clear, though in vitro studies suggest that RPA binding to the ssDNA loop could promote cleavage of the opposite strand [81,96].

Although we favor the possibility of fork convergence resolving the strand invasion intermediate, we cannot rule out a role for the SSEs, Mus81-Mms4 or Yen1, in cleavage of the D-loop to link the 3′ and 5′ ends of the *TRP1* gene flanking the region of homology shared by the repeats [97,98]. Furthermore, previous studies of ectopic recombination and targeted gene replacement suggested a role for Rad1-Rad10/XPF-ERRC1 in cleaving recombination intermediates at a heterology boundary [97–100]. Another possible mechanism following fork reversal would involve cleavage of the reversed fork by Mus81-Mms4 or Yen1, resulting in a collapsed fork. One end of the collapsed fork could then be resected to expose homology within the *trp1-3′Δ* repeat, and then this strand could invade the *5′-trp1Δ* allele of the intact sister chromatid (S6 Fig). Following DNA synthesis, the D-loop intermediate could be resolved by a converging fork, or synthesis could continue in the context of the D-loop by BIR [101,102]. Recombination by this mechanism would not generate a large loop heterology explaining the more modest defect of the *rad1Δ* mutant in generation of TDs in response to nCas9 expression. The 4-fold defect observed most likely reflects the need for Rad1-Rad10 to remove the ~138-nucleotide long 3′ heterologous flap formed following Rad51-mediated strand invasion [80]. In contrast to Tus/*Ter*-induced recombination, nCas9-induced deletions were reduced in the *rad1Δ* mutant, indicating that the mechanism of recombination at a broken replication fork is different to that at a stalled replication fork, and that fork breakage does not contribute significantly to the formation of Tus/*Ter*-induced recombinants.

The more modest defect of the *rad51Δ* mutation relative to *rad52Δ* suggests that some TDs arise by a mechanism independent of strand invasion, potentially involving strand annealing. One possible mechanism would involve convergence of a replication fork initiated from a downstream origin with the reversed fork (S7 Fig). The branched structure could be cleaved so that the duplicated region represented by the reversed fork could recombine by SSA with the broken fork. An early study of SSA in yeast showed a strong bias for use of homologies closest to the DSB ends [103], which would favor the formation of TDs over deletions in this context. A similar model of over-replication at the *RTS1* polar replication fork barrier has been proposed to explain the formation of duplication-deletion rearrangements in *S. pombe* [104].

An unequal reciprocal crossover between replicated sister chromatids could simultaneously generate duplication and deletion products [40,42]. Indeed, the suppression of both classes of recombinants by Sgs1 and requirement for HJ resolvases is consistent with a crossover model. However, the requirement for Mph1 for the majority of Tus/*Ter*-stimulated events suggests that fork remodeling is needed to provide an end for strand invasion. Since the *mph1Δ* mutant generally

exhibits a hyper-crossover phenotype in recombination assays [74,105], one would expect an increase in recombinants in the *mph1Δ* mutant if they formed by unequal exchange. The phenotype of r*ad1Δ* and *slx4Δ* mutants is also more difficult to explain by an unequal exchange mechanism.

Surprisingly, we observed that mutants lacking the Mre11 and Xrs2 components of the MRX complex display a hyper-recombination phenotype for Tus/*Ter*-induced TDs. Moreover, this hyper-recombination phenotype does not appear to depend on MRX's known roles related to non-homologous end-joining, Tel1 signaling, or replisome stability. We observed that mutants with impaired sister chromatid cohesion (*scc1-73* and *tof1Δ*) show an increased frequency of Tus/*Ter*-induced TDs, similarly to that seen in the MRX mutants. These results suggest that the increased levels of fork stalling-induced TDs could be a consequence of perturbing MRX's role in promoting sister chromatid cohesion. It is possible that sister chromatid cohesion promotes equal sister chromatid recombination at a stalled fork, and loss of cohesion could promote unequal sister chromatid recombination/misalignment between the repeats if there is more flexibility for the nascent strand to invade at an ectopic site.

Whether MRX promotes sister chromatid cohesion directly or indirectly through its role in resection is still unclear. As the Rad50 component of the MRX complex belongs to the SMC family of proteins, one possibility is that the MRX complex has a structurally similar role to the cohesin complex in promoting sister chromatid cohesion at stalled forks. A previous study has demonstrated that cohesin accumulates at stalled forks in a Rad50-dependent manner [65], suggesting that the MRX complex is important for cohesin recruitment. Our finding that overexpression of Exo1 reduces the frequency of TDs in the *mre11Δ* background to WT levels supports the possibility that the hyper-recombination phenotype of the *mre11Δ* mutant could be linked to a disruption of resection. Perhaps increased resection tract lengths behind the fork, or of a reversed fork, could promote equal SCR directly by exposing the full length of the *trp1* reporter sequence for HR, or indirectly by providing ssDNA for cohesin loading [67,106].

A previous study found that a bidirectional Tus/*Ter* block increases the frequency of microhomology-mediated TDs in mammalian cells and that BRCA1 deficiency further increases these TDs [22], suggesting that BRCA1 suppresses the formation of TDs at stalled forks. It is possible that the hyper-recombination phenotype for Tus/*Ter*-induced TDs that we observe for *mre11Δ* and *xrs2Δ* yeast strains in our study could be analogous to loss of BRCA1 in mammalian cells. BRCA1 forms a complex with CtIP and MRN during S phase [107] and has various roles in supporting end resection, particularly through preventing resection inhibition by 53BP1 [108]. The BRCA1-BARD1 complex has also been shown to promote long-range resection by EXO1 and DNA2 and may promote resection by MRN-CtIP outside of its role in 53BP1 inhibition [109,110]. BRCA1 has also been implicated in the protection of stalled forks from degradation by MRE11 [111]. Taken together with these studies, our results support the possibility that short-span TDs in cancer could arise from aberrant recombination induced by replication fork stalling, and that these events are suppressed by a resection-related mechanism.

## Materials and methods

### Yeast strains

All yeast strains are in the *S. cerevisiae* W303 background with the corrected *RAD5* allele and are listed in S4 Table. Strains were constructed by genetic crosses or integration of various cassettes with flanking homologies into the yeast genome via lithium acetate transformation. The endogenous *trp1-1* locus was replaced with a *hphMX* cassette in all strains used for recombination assays. Strains used for ChIP-qPCR also contained a *BAR1* deletion.

A direct repeat reporter consisting of two truncated *trp1* fragments, with 426 base pairs of overlapping homologies that are separated by a *Kluyveromyces lactis URA3* marker (*5'Δ-trp1::KlURA3::trp1-3'Δ*), was generated for this study. This reporter cassette was constructed by combining three PCR fragments with overlapping homologies. Specifically, the *5'Δ-trp1* and *trp1-3'Δ* fragments were amplified by PCR using pRS314 as a template and MT03 and MT04 primers for *5'Δ-trp1*, and MT05 and MT06 primers for *trp1-3'Δ*, to generate homology to chromosome VI and *KlURA3* (see S5 Table for

oligonucleotides). The *KlURA3* fragment was amplified by PCR using pOM12 as a template with primers MT07 and MT08. The three fragments were combined via PCR with primers "5′ ChrVI F" and "3′ ChrVI R." Primers "5′ ChrVI extension" and "3′ ChrVI extension" were used to extend the length of homology to Chr VI on the 5′ and 3′ ends of the combined *5′Δ-trp1::KlURA3::trp1-3′Δ*) reporterfragment. This reporter was integrated in the yeast genome 3852 bp downstream of *ARS607* at the *LSB3* locus on Chr VI. All strains were grown at 30 °C except LSY5741-19A, which was grown at the permissive temperature of 25 °C because of the temperature-sensitive *scc1-73* allele.

For Tus/*Ter* strains, the 21 *TerB* repeats (blocking orientation) were amplified by PCR from plasmid pLS616 (TMA1-21xTerB-NatMX6-TMA2) with primers oLea67-short and MT30 (S2 Table). Another round of PCR was performed on this fragment with primers MT42long and MT30 to create 57 bp homology to ChrVI upstream of the gRNA6 target site. A second fragment with homology to the 3′ end of the MT42-long+MT30 fragment and 500 bp homology to ChrVI downstream of the gRNA6 target site was created by PCR using primers MT38 and MT39. The two fragments were then assembled via another round of PCR with primers MT41 and MT39 to ultimately create a 21xTer fragment with 57 bp homology to the region upstream and 500 bp homology to the region downstream of the gRNA6 target site. This fragment was then integrated 138 bp downstream of the *trp1-3′Δ* allele via Cas9-mediated genome editing with gRNA6 [91]. The $P_{GAL1}$-*HA-Tus* cassette was integrated at the *LEU2* locus, as described previously [37].

For nCas9 strains, pLS588 was digested with AscI, and the *ARS607gRNA6-LEU2MX* fragment was integrated in yeast strains at the *leu2* locus [91]. The *lexO-Cas9^{D10A-ER}* construct was generated using the Q5 site-directed mutagenesis kit (New England Biolabs) to generate the D10A mutation in pLS642/pAA20 [90]. This plasmid, pLS749, was then digested with AscI, and the *lexO-Cas9^{D10A-ER}-HIS5MX* cassette was integrated at the *his3* locus.

## Recombination assay for detection of tandem duplications and deletions

Strains were grown for 4–5 days on synthetic complete media lacking uracil to maintain the integrity of the recombination reporter before induction of Tus. Single colonies were then picked and suspended in 1 mL of water, and cells from each colony were serially diluted and plated on YPAD (1% yeast extract, 2% bacto-peptone, 2% dextrose, 10 mg/L adenine) and either YPAG (1% yeast extract, 2% bacto-peptone, 2% galactose, 10 mg/L adenine) medium (for Tus/*Ter* strains) or YPAD medium containing a final concentration of 2 μM β-estradiol (for nCas9 strains). Strains were then grown on YPAD for 3 days and YPAG (Tus/*Ter* strains) or YPAD+2 μM β-estradiol (nCas9 strains) for 4–5 days (sometimes longer for slow-growing strains), and then individual colonies were picked from each plate and suspended in 1 mL of water. Cells were serially diluted and plated on YPAD medium, synthetic complete medium lacking tryptophan (SC-Trp), and medium containing 5-Fluoroorotic acid (5-FOA). Colonies on each plate were counted 2–3 days after plating. The frequency of Trp⁺ and Ura⁻ recombination events were determined by calculating the ratio of the number of colonies growing on SC-Trp plates or 5-FOA plates and YPAD plates. Each data point on the graphs represents the frequency of Trp⁺ or Ura⁻ recombinants measured from one colony. The median recombination frequencies are indicated on the graphs as black lines and were calculated for each strain and condition from multiple independent trials. These median values are indicated in S1-S3 Tables, in addition to the number of colonies tested for each strain. The data used to generate median recombination frequencies for all strains are provided in S6 and S7 Tables.

## Statistical analysis

Trp⁺ and Ura⁻ recombination frequencies were analyzed on log-transformed values by one-way ANOVA with a Bonferroni post-test. Spontaneous and replication stress-associated data were analyzed separately. For single mutants, asterisks indicate a significant difference with the WT strain in the same growth condition: ns (not significant) $p > 0.05$, \*p-value<0.05, \*\*p-value<0.005, \*\*\*p-value<0.001, \*\*\*\*p-value<0.0001. For double mutants, comparisons were made with their respective single mutants.

## Southern blotting

Genomic DNA was extracted from Trp+ and Ura- colonies post-recombination assay, as well as from Trp-Ura+ cells prior to performing the recombination assay, using the MasterPure Yeast DNA Purification kit (Biosearch Technologies). DNA concentrations were measured using the Qubit Flex fluorometer and 1 x HS dsDNA assay kit (Invitrogen). 1 µg genomic DNA was digested overnight with PvuII-HF in rCutsmart buffer (New England Biolabs) in a 30 µL reaction and subsequently run on a 0.8% agarose gel made with 1 X TBE. After depurination, denaturing, and neutralization of the gel, digestion products were transferred overnight in 2 X SSC from the gel to a Hybond N+ membrane (Cytiva/Amersham). The membrane was then UV-crosslinked at 1200 J and hybridized using a $^{32}$P dCTP-labeled *TRP1* probe in ULTRA-hyb Ultrasensitive hybridization buffer (Invitrogen) and exposed to a phosphorimager screen and imaged.

## Spot assays to assess sensitivity to replication stress

Single colonies were inoculated in 5 mL liquid YPAD media overnight at 30 °C. The next day, cultures were diluted to an $OD_{600}$ of 0.3, and tenfold serial dilutions of these cells were spotted onto YPAD media and YPAG (for Tus/*Ter* strains) or YPAD+2 µM β-estradiol (for nCas9 strains) media. Cells were grown at 30°C for 2–5 days and then imaged.

## Chromatin immunoprecipitation-qPCR for Mcm2–7

Overnight cultures of single colonies were grown in 5 mL YP medium containing 2% glucose (YPAD). In the morning, 500 µL cells were diluted in 50 mL YP medium containing 2% raffinose (YPAR) and allowed to grow overnight again. The next day, cells were then diluted to $OD_{600}$ of 0.1 and grown to an $OD_{600}$ of 0.3-0.5. Cells were then arrested in G1 with 50 ng/mL α-factor for 3.5 hrs. Galactose was added to a concentration of 2% after one hour. Medium was then removed, and cells were washed twice with water containing 50 µg/mL pronase. Cells were released into YPAR medium with (+Tus) or without (-Tus) 2% galactose and 100 µg/mL pronase, and timepoints were collected at different times following release (0 min., 35 min., 45 min., 55 min.). At each timepoint, cells were crosslinked with 1% formaldehyde for 15 min. and then quenched with 125 mM glycine for 5 min. Cells were washed with HBS (50 mM HEPES pH 7.5, 140 mM NaCl) and then with ChIP lysis buffer (50 mM HEPES pH 7.5, 140 mM NaCl, 1 mM EDTA, 1% IGEPAL CA-630, 0.1% deoxycholate). Cells were then lysed with acid-washed glass beads in fresh ChIP lysis buffer containing 1 Roche mini protease inhibitor tablet/10 mL buffer and 1 mM PMSF using a FastPrep-24 (MP Biomedicals). Cells were then further lysed using a QSonica Q800R sonicator (10 cycles at Amp. 65% for 30 seconds ON, 45 seconds OFF). Chromatin immunoprecipitation was performed using Pierce Protein A/G Magnetic Beads (Thermo Scientific) that had been incubated overnight with antibody UM174 (1.8 µL antibody per 450 uL ChIP sample) [58,59]. qPCR was performed using SsoAdvanced Universal SYBR Green Supermix (Bio-Rad). Primers ChIP-qPCR-F1 and ChIP-qPCR-R1 were used to amplify the 138 bp region between the end of the *3'Δ-trp1* fragment and the first *TerB* repeat. Percent input was calculated using the following formula: *% input = 2^((Cq^Input-6.644)-Cq^IP)\*100.*

## Fluorescence-activated cell sorting (FACS)

For each sample, $1 \times 10^7$ cells were collected by centrifugation at 2,000 x g for 3 minutes. Cells were then fixed in 70% ethanol overnight at 4 °C. Cells were then resuspended in 500 µL of 50 mM Tris-HCl pH 7.5 and incubated at room temperature for 10 minutes, and this process was repeated a second time. Cells were then resuspended in 500 µL RNase A (1 mg/mL) in 50 mM Tris-HCl pH 7.5 and incubated at 37 °C overnight. The following day, cells were spun down at 2,000 x g for 3 minutes and then treated with 500 µL Proteinase K (1 mg/mL) in 50 mM Tris-HCl pH 7.5 and incubated at 50 °C for 1 hour. After resuspension in 500 µL 200 mM Tris-HCl pH 7.5 200 mM NaCl 80 mM $MgCl_2$, 100 µL cells were diluted in 1 mL of SYTOX Green (Invitrogen) to a final concentration of 300 nM. Cells were then sonicated and analyzed on a BD LSR II Flow Cytometer.

## Supporting information

**S1 Fig. Yeast strains tested in this study are not sensitive to the Tus/*Ter* block.** Tenfold serial dilutions of selected strains containing the direct repeat reporter along with the galactose-inducible Tus/*Ter* system plated on YPAD (-Tus) or YPGAL (+Tus) media and grown for 2 days (YPAD) or 3 days (YPGAL).
(TIF)

**S2 Fig. Loss of both Rad51 and Rad59 abolishes TDs induced by a Tus/*Ter* block.** Frequency of Trp+ (A) and Ura- (B) recombinants in WT, *rad51Δ*, *rad59Δ*, and *rad51Δ rad59Δ* strains. Statistical significance was determined by one-way ANOVA on log-transformed data with a Bonferroni post-test. p-values are indicated as follows: ns (not significant) $p > 0.05$, *$p < 0.05$, **$p < 0.005$, ***$p < 0.001$, ****$p < 0.0001$.
(TIF)

**S3 Fig. Loss of Mre11 does not affect replisome stability at the Tus/*Ter* barrier.** A. FACS profiles for WT and *mre11Δ* cells after release from G1 arrest -/+Tus expression. B. ChIP-qPCR for Mcm2–7 using primers 138 bp upstream of the *Ter* repeats.
(TIF)

**S4. Fig. Epistasis between *rad1Δ* and *slx4Δ* mutations.** Frequency of Trp+ (A) and Ura- (B) recombinants in WT, *rad1Δ*, *slx4Δ*, and *rad1Δ slx4Δ* strains. Statistical significance was determined by one-way ANOVA on log-transformed data with a Bonferroni post-test. p-values are indicated as follows: ns (not significant) $p > 0.05$, *$p < 0.05$, **$p < 0.005$, ***$p < 0.001$, ****$p < 0.0001$.
(TIF)

**S5 Fig. Characterization of the nCas9 system.** A. Frequencies of Trp+ recombinants in WT strains containing the direct repeat reporter in the absence of replication stress, with the Tus/*Ter* system, or with the Cas9^D10A/gRNA6. B. Ten-fold serial dilutions of the indicated strains with Cas9^D10A/gRNA6 plated on YPAD (-nCas9) or YPD+β-estradiol (+ nCas9) media and grown for 2 days (YPAD) or 3 days (YPD+β-estradiol).
(TIF)

**S6 Fig. Model for TD formation at a broken replication fork.** Cleavage of the reversed fork or nicking of the leading strand template downstream of direct repeats would generate a single-ended DSB. Exo1 degradation of the broken arm would create ssDNA for Rad51 assembly and strand invasion. Invasion of the *5′Δ-trp1* repeat by *trp1–3′Δ*, followed by DNA repair synthesis in the context of a migrating D-loop would reconstitute a functional *TRP1* gene (TD). Invasion of the *trp1–3′Δ*, repeat by *5′Δ-trp1* would have the potential to generate a deletion product. The D-loop could be resolved by an incoming replication fork or BIR to the telomere. The region of homology shared by the repeats is shown in mid-blue, while the 3′ end and 5′ end of *TRP1* are shown in dark blue. Parental strand is indicated by black lines, nascent strand by gray lines, and repair-associated DNA synthesis by a dashed gray line.
(TIF)

**S7 Fig. Model for TD formation by SSA.** Upon encountering a Tus/*Ter* barrier, the replication fork could undergo reversal mediated by Mph1. The reversed fork could then be resected through the nuclease activity of Exo1. Alternatively, Exo1 degradation of the nascent lagging strand at the stalled fork would create a ssDNA gap to facilitate pairing of the parental strands, displacing the nascent leading strand. An incoming replication fork could stall at the reversed fork with the over-replicated arm. Cleavage of stalled fork would create a break with overlapping regions of homology to the region of over-replication. Resection of the lagging strand of the broken fork followed by strand annealing would reconstitute a functional copy of *TRP1* (TD). The region of homology shared by the repeats is shown in mid-blue, while the 3′ end and 5′

end of *TRP1* are shown in dark blue. Parental strand is indicated by black lines, nascent strand by gray lines, and repair-associated DNA synthesis by a dashed gray line.
(TIF)

**S1 Table. Median Trp$^+$ recombination frequencies of strains with a Tus/*Ter* block.**
(DOCX)

**S2 Table. Median Ura$^-$ recombination frequencies of strains with a Tus/*Ter* block.**
(DOCX)

**S3 Table. Median Cas9$^{D10A}$-induced recombination frequencies.**
(DOCX)

**S4 Table. Yeast strains used in this study.** All strains are in the W303 background (*leu2–3,112 ura2–1 ade2–1 his3–11,15 can1–100*).
(DOCX)

**S5 Table. Oligonucleotides used in this study.**
(DOCX)

**S6 Table. Raw data used for generation of graphs in Figs 1–5, S2 and S4.**
(XLSX)

**S7 Table. Raw data used for generation of graphs in Figs 6 and S5.**
(XLSX)

## Acknowledgments

We thank S. Zha, A. Ciccia, L. Berchowitz, W.K. Holloman and the members of the Symington lab for their helpful discussions, A. Sane for assistance with FACS, S. Bell for the gift of the anti-Mcm2–7 antibodies and R. Rothstein for gifts of yeast strains.

## Author contributions

**Conceptualization:** Marina K Triplett, Lorraine S. Symington.

**Data curation:** Marina K Triplett, Iffat Ahmed, Swathi Shekharan, Lorraine S. Symington.

**Formal analysis:** Marina K Triplett, Lorraine S. Symington.

**Funding acquisition:** Marina K Triplett, Lorraine S. Symington.

**Investigation:** Marina K Triplett, Iffat Ahmed.

**Methodology:** Iffat Ahmed, Swathi Shekharan, Lorraine S. Symington.

**Project administration:** Lorraine S. Symington.

**Resources:** Lorraine S. Symington.

**Supervision:** Lorraine S. Symington.

**Validation:** Lorraine S. Symington.

**Visualization:** Marina K Triplett, Lorraine S. Symington.

**Writing – original draft:** Marina K Triplett, Lorraine S. Symington.

**Writing – review & editing:** Lorraine S. Symington.

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
