## [Decision Letter · Decision Letter 0]

24 Mar 2025

PGENETICS-D-25-00176

The Slx4-Rad1-Rad10 nuclease differentially regulates deletions and duplications induced by a replication fork barrier

PLOS Genetics

Dear Dr. Symington,

Thank you for submitting your manuscript to PLOS Genetics. After careful consideration, we feel that it has merit but does not fully meet PLOS Genetics's publication criteria as it currently stands. Therefore, we invite you to submit a revised version of the manuscript that addresses the points raised during the review process.

Please submit your revised manuscript within 30 days Apr 23 2025 11:59PM. If you will need more time than this to complete your revisions, please reply to this message or contact the journal office at plosgenetics@plos.org. Please include the following items when submitting your revised manuscript:

We look forward to receiving your revised manuscript.

Kind regards,

Ashok Bhagwat, Ph.D.

Academic Editor

PLOS Genetics

Giovanni Bosco

Section Editor

PLOS Genetics

Aimée Dudley

Editor-in-Chief

PLOS Genetics

Anne Goriely

Editor-in-Chief

PLOS Genetics

**Additional Editor Comments :**

I am delighted to see that all three reviewers were positive about the experimental design and found the conclusions you reached in the manuscript to be important and convincing. They judged the quality of the work to be high. However, two of the reviewers did raise questions about several important scientific issues, statistical validity of the data and the presentation of data in figures. I would strongly urge the authors to respond to these questions and suggestions to improve their manuscript.

**Journal Requirements:**

At this stage, the following Authors/Authors require contributions: Swathi Shekharan, and Lorraine S. Symington. Please ensure that the full contributions of each author are acknowledged in the "Add/Edit/Remove Authors" section of our submission form.

The list of CRediT author contributions may be found here: https://journals.plos.org/plosgenetics/s/authorship#loc-author-contributions

https://journals.plos.org/plosgenetics/s/submission-guidelines#loc-parts-of-a-submission

5) We notice that your supplementary Figures, and Tables are included in the manuscript file. Please remove them and upload them with the file type 'Supporting Information'. Please ensure that each Supporting Information file has a legend listed in the manuscript after the references list.

Potential Copyright Issues:

i) Figures 2A, 7, S5, and S6. Please confirm whether you drew the images / clip-art within the figure panels by hand. If you did not draw the images, please provide (a) a link to the source of the images or icons and their license / terms of use; or (b) written permission from the copyright holder to publish the images or icons under our CC BY 4.0 license. Alternatively, you may replace the images with open source alternatives. See these open source resources you may use to replace images / clip-art:

7) Please amend your detailed Financial Disclosure statement. This is published with the article. It must therefore be completed in full sentences and contain the exact wording you wish to be published.

2) If any authors received a salary from any of your funders, please state which authors and which funders

8) Please ensure that the funders and grant numbers match between the Financial Disclosure field and the Funding Information tab in your submission form. Note that the funders must be provided in the same order in both places as well. Currently, the order of the grants is different in both places.

**Reviewers' comments:**

Reviewer's Responses to Questions

Reviewer #1: The Symington lab has carried out an excellent study of copy number variation induced by a conditional replication fork block. The mechanism(s) by which deletions and tandem duplications arise appear to have several genetic distinctions, which likely rules out that most of the events are not simply the result of an unequal sister chromatid exchange (but this could be made more clear). There are many factors at play: Mre11, Exo1, cohesin, but the authors emphasize the role of Slx4 and the Rad1-Rad10 nuclease that appears here to prevent the deletion outcomes. The authors also find that a Cas9-directed nick has outcomes distinctly different from the fork block, most notably with Rad1. Overall the data are clearly presented, though I had a question about the statistical method used.

Other concerns are listed here

l. 130. How large is each Ter repeat? And – though I am not asking for more experiments – is there any insight as to how close to the Ter repeats the stalling occurs and what effect the distance to the recombination construct might have?

It isn’t clear to me that the fork will be blocked only after the reporter 3’ TRP-URA3-5’ TRP construct has been fully copied. If the block happens sooner, so that perhaps only the 3’ TRP portion has been copied, how might that affect the ratio of outcomes?

l. 170. Is there any cell cycle delay caused by the stalled fork? Fig. S3 doesn’t show later time points where a delay might be evident.

l. 253. How is the scc1 mutant conditional and how was it assayed?

Figures: I am unclear what a Bonferroni “post test” is, or how p<0.05 would still be significant in the case of multiple comparisons. I thought the criteria for significance should drop to a lower p value.

Fig. 6. A. I thought that a structure-specific endonuclease would convert the reversed fork to a one-ended structure but with a light blue double-stranded end (yielding a structure similar to that shown in S5). I found this figure confusing. Maybe one needs to distinguish between HJ-cleaving SSEs and one that cleave a single strand (Rad1-10 and others).

Fig. S5. Could this replication require BIR factors?

Fig. 7. It took quite a while for me to understand Fig. 7, which is the crux of the paper. First of all, the step shown below the labels (e.g. “removal of 3’ heterologous end”) are actually the substrates for that specific step, rather than the outcome. Towards the end of the Figure there should be an “and” inserted to make it clear these are the two outcomes of a single replication event.

Although it will make the figure a bit more dense, I think the TRP-URA3 repeats have to be shown instead of the faint gray loop. On the right side of the Figure, the invasion is particularly confusing: the gray line has (I think) zero length and the invasion actually creates a dsDNA loop adjacent to a D-loop. How this gets resolved is not clear to me.

Also, the deletion outcome on the right side is missing.

Moreover, whereas the bottom of the right side shows Rad1-10 is needed to cut the heteroduplex loop at its base (and hence when this doesn’t happen, deletions would go up - presumably as a sectored colony), BUT, Rad1-10 is also invoked above, to create the loop in the first place. So, if Rad1-10 is needed here, how can that explain the increase in deletions? This seems to be a problem, but I may have missed the way to resolve it.

General: simple unequal crossing-over could generate a TD and a deletion simultaneously. This possibility doesn’t get much attention, though I understand that the different effects of various mutants rule out that the majority of events occur in this way. Long-tract GC is also introduced, but again, whether some of these events arise in that way isn’t dealt with except to indicate it could happen and might occur more without sister-chromatid cohesion. LTGC is a special form of BIR. As noted above, it might be good to see how pif1 or pol32 affect these outcomes.

Reviewer #2: Triplett et al have developed a clever in vivo system for characterizing repeat instability in response to replication stress. The replication stress is induced by induced Tus protein binding to an array of Ter sites downstream of the direct repeats. The system is able to detect both duplications (Trp+) and deletions (Ura-) that occur without and with Tus induction, via galactose induction of a pGAL-HA-Tus cassette. The authors generated mutations in several genetic pathways involved in processing replication/recombination intermediates, to identify factors that enhance or suppress duplications and/or deletions following the Tus-mediated replication stress. They found that the Tus/Ter-induced recombination is RAD51- and RAD52-dependent, indicating the generation of strand invasion recombination intermediates. Loss of MRE11 or XRS2 increased recombination, in a RAD52-dependent manner, likely independent of its role in resection (MRE11 nuclease activity not required), in NHEJ or checkpoint activation. scc1-73 and loss of TOF1, both of which compromise sister chromatid cohesion, also increased recombination. Loss of EXO1 reduced recombination, likely due to loss of long range resection, while loss of SGS1 increased recombination. The authors suggest that this is due to the loss of Sgs1 recombination dissolution activity. Loss of MPH1 also reduced recombination, potentially affecting fork reversal and/or D-loop dissociation. Finally, loss of either SLX4 or RAD1 reduced duplication recombinants but increased deletions. These were the only factors that had opposite effects on duplications and deletions. Loss of RAD1 and RAD51 or RAD52 resulted in synergistic decreases in duplication recombinants and the RAD1-dependent hyper-deletion phenotype was RAD51- and RAD52-dependent. The authors tested the effect of fork breakage at the direct repeats on the genetic requirements for duplications and deletions and found that MPH1, MUS81 and YEN1 are dispensable for tandem duplications, in contrast to what was observed with the Tus/Ter system. In this modified system, RAD1 was required for deletion events, again in contrast to the TUS/Ter system. These results provide insights into the formation and processing of different recombination intermediates during replication stress (fork stalls). The authors propose a model for copy number variation (duplication or deletion) at stalled replication forks, highlighting the potential role of RAD1 in promoting duplications and preventing deletions. They also propose models for tandem duplication a broken replication forks (Cas9) and through single strand annealing.

1. The data are convincing and the experiments are carefully done and controlled. My major question is about the role of MSH3 or SAW1, both of which function with Rad1-Rad10 in processing 3’ non-homologous tails in recombination intermediates, through recruitment to 3’ ssDNA flaps. The repeat lengths used here are within the range that would show a Msh2-Msh3 effect in SSA. MSH3 has also been shown to be involved in the large loop repair attributed to Rad1-Rad10 in the models here. Msh2-Msh3 also recruits Rad1-Rad10 to B-/Z-DNA junctions. Understanding the contribution of MSH3 and SAW1 would help support and refine the proposed models.

2. Further, Msh2-Msh3 interacts with Sgs1 to recruit it to recombination intermediates with homeologous sequences. MSH2 has also been shown to modulate sister chromatid exchange in the presence of inverted repeats. Increased unequal sister chromosome exchange in SGS1 mutations was partially MSH2-depenednt. Again, the role, if any, for MSH proteins can help refine mechanism and models.

3. Related – the model appears to be that Slx4 recruits Rad1-Rad10 to the recombination intermediates for processing. Are the rad1Δ and slx4Δ epistatic or synergistic for the effect on duplications or deletions? Are the nCas9-induced deletions also dependent on Slx4, as well as Rad1-Rad10.

Reviewer #3: This is an excellent paper dedicated to the mechanisms responsible for the copy number variations (CNV) that are of pivotal importance for genomic instability in health and disease. The authors attempted to unravel how replication stress leads to CNV. To this end, they developed an elegant reporter cassette to study genetic controls of CNV upon replication stress in a yeast S. cerevisiae. Specifically, they analyzed recombinational mechanisms leading to duplications or deletions upon replication fork stalling at Tus/Ter complex. Their genetic analysis identified the Mph1 translocase, Exo1 exonuclease, as well as Rad52 & Rad51 recombinases as the key proteins driving CNV formation. Remarkably, Slx4 scaffold protein and Rad1-Rad10 nucleases promoted duplications, but counteracted deletion formation. Altogether, these data led them to propose a new and thought-provoking model of how the repair of the stalled replication fork leads to CNV. First, locally stalled replication fork undergoes reversal driven by the Mph1 translocase, which is followed by Exo1 end-resection and strand invasion driven by Rad52 and Rad51. Second, DNA synthesis at the resultant D-loop followed by its subsequent resolution by the convergent replication fork leads to the formation of a specific heteroduplex intermediate flanked by the large loop. Third, resolution of this loop by the Rad1-Rad10 complex promotes duplications while eliminates deletions.

The experimental design is excellent, the results are clear-cut and their interpretations are sound. Nothing to add or subtract. I was particularly impressed by their model involving large-loop heteroduplex intermediates, which made me to rethink some of my own data.

**Have all data underlying the figures and results presented in the manuscript been provided?**

Reviewer #1: Yes

Reviewer #2: Yes

Reviewer #3: Yes

PLOS authors have the option to publish the peer review history of their article (what does this mean? ). If published, this will include your full peer review and any attached files.

**Do you want your identity to be public for this peer review?** For information about this choice, including consent withdrawal, please see our Privacy Policy .

Reviewer #1: **Yes: ** James E. Haber

Reviewer #2: No

Reviewer #3: **Yes: ** Sergei Mirkin

**Figure resubmission:**
---

## [Editor Report · Decision Letter 1]

10 May 2025

Dear Dr Symington,

We are pleased to inform you that your manuscript entitled "The Slx4-Rad1-Rad10 nuclease differentially regulates deletions and duplications induced by a replication fork barrier" has been editorially accepted for publication in PLOS Genetics. Congratulations!

Yours sincerely,

Ashok Bhagwat, Ph.D.

Academic Editor

PLOS Genetics

Giovanni Bosco

Section Editor

PLOS Genetics

Aimée Dudley

Editor-in-Chief

PLOS Genetics

Anne Goriely

Editor-in-Chief

PLOS Genetics

Comments from the reviewers (if applicable):

**Data Deposition**

http://datadryad.org/submit?journalID=pgenetics&manu=PGENETICS-D-25-00176R1

**Press Queries**

---

## [Editor Report · Acceptance letter]

PGENETICS-D-25-00176R1

The Slx4-Rad1-Rad10 nuclease differentially regulates deletions and duplications induced by a replication fork barrier

Dear Dr Symington,

We are pleased to inform you that your manuscript entitled "The Slx4-Rad1-Rad10 nuclease differentially regulates deletions and duplications induced by a replication fork barrier" has been formally accepted for publication in PLOS Genetics! Your manuscript is now with our production department and you will be notified of the publication date in due course.

With kind regards,

Lilla Horvath

PLOS Genetics

On behalf of:
